# The Significant Decrease of Serum Sodium and Blood Pressure following Thoracoscopic Left Atrial Appendage Clipping

**Yiming Chen, Xuesong Han, Cong Ye and Dong Xu \***

Department of Cardiac Surgery, Beijing Tiantan Hospital, Capital Medical University, Beijing 100071, China
*   Correspondence: dr_dxu@163.com

**Abstract:** Background: The epicardial left atrial appendage (LAA) closure may induce unwanted natriuretic peptides caused by the isolation of the LAA from circulation. Thus, this study aims to explore the possible change of blood pressure and electrolytes following the procedure. Methods: This was a retrospective, observational study including 52 atrial fibrillation (AF) patients with a history of thrombolic stroke who underwent thoracoscopic LAA clipping. Electrolytes, blood pressure, and brain natriuretic peptide were measured before the procedure, immediately after the device release, on the 1st day, the 2nd day postoperation, and discharge. Results: Thirty-five (66.04%) patients' serum sodium level decreased by more than 4 mmol/L during 48 h postoperation. The systolic blood pressure at discharge was significantly lower than the baseline level (118.99 ± 12.29 mmHg vs. 122.93 ± 13.82 mmHg, $p = 0.034$), while the diastolic blood pressure was not significantly different to the baseline (78.00 ± 7.39 mmHg vs. 77.22 ± 7.72 mmHg, $p = 0.502$). A significant increase in brain natriuretic peptide was observed postoperatively, although it showed a trend of decline at discharge. Conclusion: Epicardial LAA clipping could induce an acute decrease in serum sodium postoperation, which indicates to the surgeons that the postoperative intake fluid amounts and serum sodium level management should be more appropriate. The decrease in systolic blood pressure indicates the possibility of expanding the benefits that patients received from LAA clipping, though further studies are needed to determine if this phenomenon is persistent in the long-term follow-up.

**Keywords:** atrial fibrillation; stroke; natriuretic peptide





## 1. Introduction

Atrial fibrillation (AF) is the most common type of tachyarrhythmia worldwide [1]. Approximately 90% of thrombi in patients with nonvalvular AF (NVAF) are oriented from the left atrial appendage (LAA) [2], and the risk of stroke is five times higher than normal [3]. Procedures aimed at preventing cardiogenic stroke by removing the LAA from circulation have been used for decades. As the derivative of the primitive fetal atrium [4], LAA is important not only for mechanical circulation but also for its neurohormonal function. LAA is the well-known source of atrial natriuretic peptide (ANP) and brain natriuretic peptide (BNP), both of which are produced in the cardiocytes and secreted into the circulation to affect the patients' systemic homeostasis [5–7]. Because of the neurohormonal function of LAA mentioned above, the removal of LAA may cause electrolyte and hemodynamic changes in patients with NVAF.

Maybrook et al. [8] discovered that patients who underwent epicardial LAA closure with a LARIAT suture delivery device experienced an early and sustained drop in systolic blood pressure, while the serum sodium also dropped early but returned to the baseline during long-term follow-up. Lakkireddy et al. [9] confirmed this phenomenon and reported that it was not seen in endocardial LAA occlusion through the WATCHMAN. The percutaneous LAA closure through the LARIAT device could induce the in situ necrosis of LAA, as well as the procedure performed by the E-clip (Med-zenith, Beijing, China) device. Thus,

both of these procedures were generally identified as epicardial LAA closure; however, we still lack an understanding of the change of homeostasis among AF patients with a history of thrombolic stroke following the procedure performed by a novel LAA clipping device. Furthermore, in previous studies, more attention was focused on the long-term results, and the specific change of serum electrolytes during hospitalization is vague. Thus, the goal of this study is to investigate the change of electrolytes and blood pressure in AF patients following thoracoscopic LAA clipping surgery.

## 2. Materials and Methods

### 2.1. Patients Selection

This is a retrospective, observational study of 52 patients with AF who underwent a successful LAA clipping procedure with a E-clip (Med-zenith, Beijing, China) in our institution between September 2019 and July 2022. If the patients met the following inclusion criteria, the procedure was carried out: (1) age $\geq$ 18 years, (2) nonvalvular AF, (3) at least one risk factor for embolic stroke (CHA2DS2-VASc score $\geq$ 1), and (4) a poor candidate or unwilling to have long-term oral anticoagulation (OAC) therapy. The procedure would not be performed if any of the following exclusion criteria were met by the patients: (1) a history of pericarditis, (2) a history of cardiac surgery, (3) severe pectus excavatum, (4) a history of the hemodynamically valvular disorder, (5) an embolic event within the last 30 days, (6) preoperative renal dysfunction, (7) New York Heart Association class III to IV heart failure symptoms, (8) left ventricular ejection function less than 30%, (9) myocardial infarction within the last 3 months, and (10) a history of thoracic radiation therapy. In addition, patients had to meet the following criteria to participate in the study: (1) Complete pre-and postoperative electrolyte and hemodynamic data, (2) no other cardiac surgeries were performed at the same time, and (3) no need for special fluid management or diuretics during the perioperative period.

### 2.2. Data Collection

Our institutional patient registry was used to obtain baseline characteristics. Electrolytes (serum sodium, potassium), hemodynamics (systolic blood pressure and diastolic blood pressure), and BNP levels were assessed preoperatively (the data of electrolytes and BNP were generally collected 1–2 days after hospitalization, and the data of BP were generally collected 1–2 days before the procedure), immediately postoperatively, on the 1st day postoperative, on the 2nd day postoperative, and before discharge. We measured the heart rate and blood pressure four times within a 24 h per time point, then averaged the results.

### 2.3. Stand-Alone Thoracoscopic Left Atrial Appendage Clipping

The procedure was carried out using a new epicardial LAA clipping system called E-lip (Med-zenith, Beijing, China), which consists of a self-closing clip made of two parallel titanium tubes with elastic nitinol springs covered by braided polyester. Preoperatively, a transoesophageal echocardiography (TEE) probe was introduced to ensure the absence of thrombi. All patients were placed supine, given general anesthesia, and intubated with a double-lumen endotracheal tube. A standard left-sided minimally invasive thoracoscopic approach was used to successfully introduce the working instruments and implant the device, followed by a posterior pericardiotomy to expose LAA. The ruler was used intra-operatively to measure the base of the LAA and size it for the appropriate E-clip length (either 35, 40, 45, or 50 mm). Before the clip was released, the device's position would be confirmed using thoracoscopic vision. If a residual LAA flow or stump was detected, the device would be repositioned. Following the procedure, a chest tube was inserted through one of the ports into the left-sided costodiaphragmatic recess.

### 2.4. Postoperative Care

Patients will be monitored in the cardiac care unit, extubated the same day, and transferred to the ward the following. Following the procedure, oral anticoagulation and antiplatelet therapy were discontinued.

### 2.5. Statistical Analysis

Normal distribution, abnormal distribution continuous, and categorical variables were expressed as mean ± SD, median (IQR), and counts (percentage), respectively. A paired *t*-test was used to compare the baseline continuous variables to those at various postprocedure time points, and 2-tailed $p < 0.05$ was considered statistically significant. SPSS Version 26 (SPSS Inc., Chicago, IL, USA; released 2013) was used for statistical analysis.

## 3. Results

### 3.1. Patient Characteristics

The study included 52 patients who underwent stand-alone thoracoscopic LAA clipping. Table 1 displays the baseline characteristics. The majority of patients were male (*N* = 42, 80.77%), and there was a mean age of 68.50 (6.00) for the entire cohort. Patients with paroxysmal AF accounted for 20.08% (*N* = 12) of all patients. All of the patients had a preoperative stroke. The average CHA2DS2-VASc score was 5.00 (1.75). Patients with hypertension accounted for 82.69% (*N* = 43) of all patients.

**Table 1.** Baseline clinical characteristics.

| | |
|---|---|
| Age | 68.50 (6.00) |
| Male | 42 (80.77%) |
| Type of Atrial fibrillation | |
|     Paroxysmal | 12 (20.08%) |
|     Non-Paroxysmal | 40 (76.92%) |
| Stroke History | 52 (100.00%) |
| Hypertension | 43 (82.69%) |
| DM | 17 (32.69%) |
| CVA/TIA | 52 (100.00%) |
| Heart failure | 4 (7.69%) |
| CHA2DS2-VASc Score | 5.00 (1.75) |
| Serum Level of Electrolytes | |
|     Sodium (mmol/L) | 142.22 ± 2.59 |
|     Potassium (mmol/L) | 3.89 ± 0.37 |
| BNP (mmol/L) | 243.11 ± 233.47 |
| Blood Pressure | |
|     SBP (mmHg) | 122.93 ± 13.82 |
|     DBP (mmHg) | 77.22 ± 7.72 |

DM: diabetes mellitus; CVA: cerebrovascular accident; TIA: transient ischemic attacks; SBP: systolic blood pressure; and DBP: diastolic blood pressure; BNP: brain natriuretic peptide.

### 3.2. Changes of Electrolytes and Hemodynamics

Table 2 displays the result of electrolytes, blood pressure, and BNP data collected at the predetermined time intervals. The baseline level of serum sodium was 142.22 ± 2.59 mmol/L (Figure 1) and decreased immediately after the procedure (139.07 ± 2.36 mmol/L, $p < 0.001$), and this decrease persisted to the discharge. A significant increase in potassium was observed immediately postoperation; then, the potassium level showed a decreasing trend up to discharge, and there was no significant difference at discharge when compared with the baseline level.

**Table 2.** Electrolyte and hemodynamic data.

| | Baseline | Immediately | 1st Day Post | 2nd Day Post | Discharge |
|---|---|---|---|---|---|
| Electrolytes | | | | | |
| Na$^+$ (mmol/L) | 142.22 ± 2.59 | 139.07 ± 2.36 ($p < 0.001$) | 140.03 ± 2.84 ($p < 0.001$) | 138.47 ± 3.10 ($p < 0.001$) | 138.48 ± 2.63 ($p < 0.001$) |
| K$^+$ (mmol/L) | 3.89 ± 0.37 | 4.11 ± 0.49 ($p = 0.005$) | 4.33 ± 1.01 ($p = 0.005$) | 4.23 ± 0.49 ($p < 0.001$) | 3.98 ± 0.39 ($p = 0.149$) |
| Blood Pressure | | | | | |
| SBP (mmHg) | 122.93 ± 13.82 | 131.94 ± 12.93 ($p < 0.001$) | 120.14 ± 10.68 ($p = 0.193$) | 119.12 ± 11.81 ($p = 0.117$) | 118.99 ± 12.29 ($p = 0.034$) |
| DBP (mmHg) | 77.22 ± 7.72 | 75.32 ± 7.59 ($p = 0.218$) | 66.35 ± 8.44 ($p < 0.001$) | 71.26 ± 10.31 ($p = 0.001$) | 78.00 ± 7.39 ($p = 0.502$) |
| BNP (pg/mL) | 174.90 (239.70) | 236.70 (207.30) ($p = 0.030$) | 314.00 (260.90) ($p < 0.001$) | 398.50 (337.70) ($p < 0.001$) | 227.90 (305.40) ($p = 0.035$) |

*p* values are relative to the baseline values.

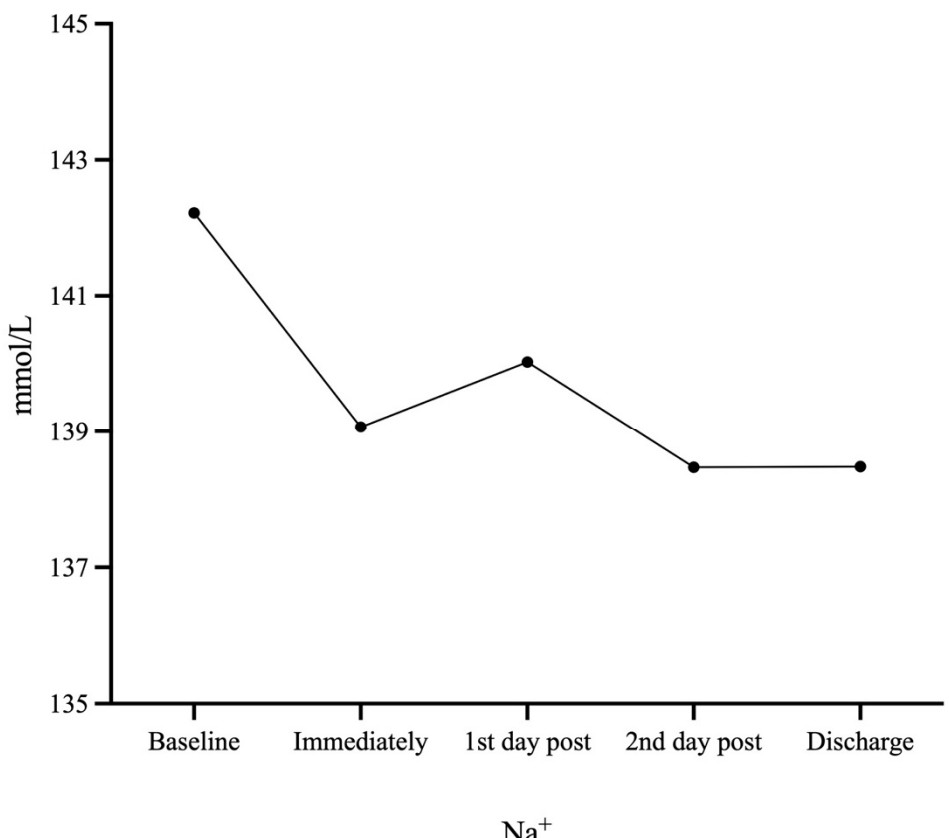

Na$^+$

**Figure 1.** The change of blood pressure postoperation.

The systolic blood pressure (SBP) increased immediately after the operation when compared with baseline (131.94 ± 12.93 vs. 122.93 ±13.82, *p* < 0.001), then decreased to 120.14 ± 10.68 and 119.12 ± 11.81 on the 1st and 2nd day postoperative, respectively. However, when compared with the baseline, neither time point was statistically significant. Before discharge, the mean SBP was still lower than the baseline with a significant decrease (118.99 ± 12.29 vs. 122.93 ± 13.82, *p* = 0.034). The significant decline of diastolic blood pressure (DBP) was observed on the 1st and 2nd day postoperative; however, it returned to the baseline at hospital discharge (Figure 2).

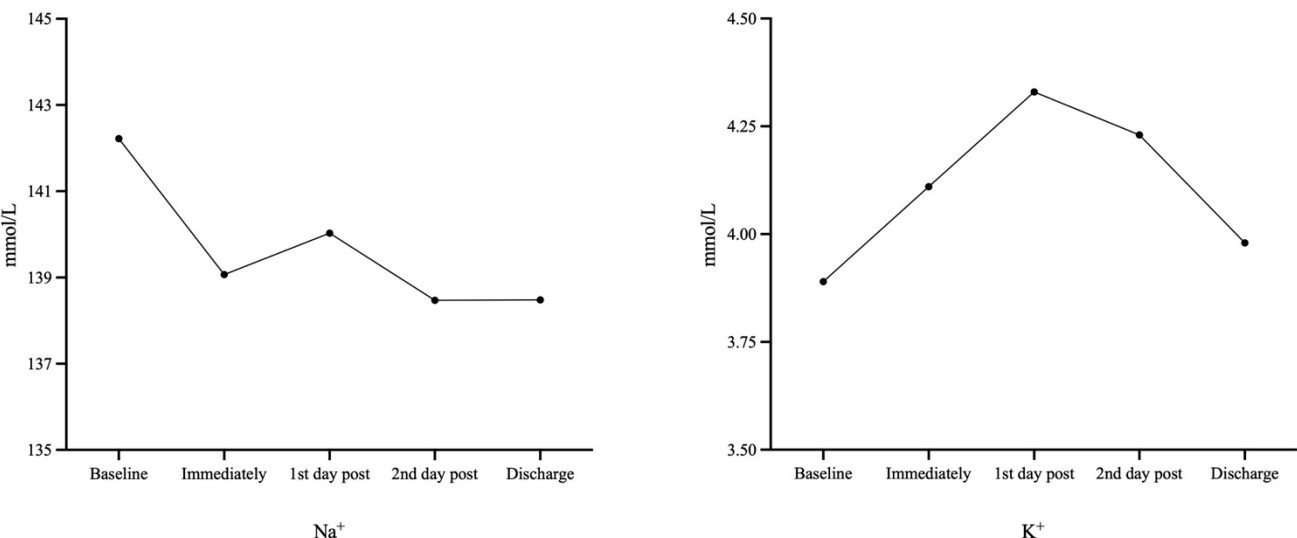

**Figure 2.** The change of serum electrolytes postoperation.

*3.3. Change of Brain Natriuretic Peptide*

The baseline level of BNP was 174.90 (239.70) pg/mL, and a significant increase was observed immediately postoperation. Meanwhile, this increase was persistent to discharge. Although BNP showed a recovery trend at discharge, it was still significantly higher than the baseline level (227.90 (305.40) pg/mL vs. 174.90 (239.70) pg/mL, $p = 0.035$) (Figure 3).

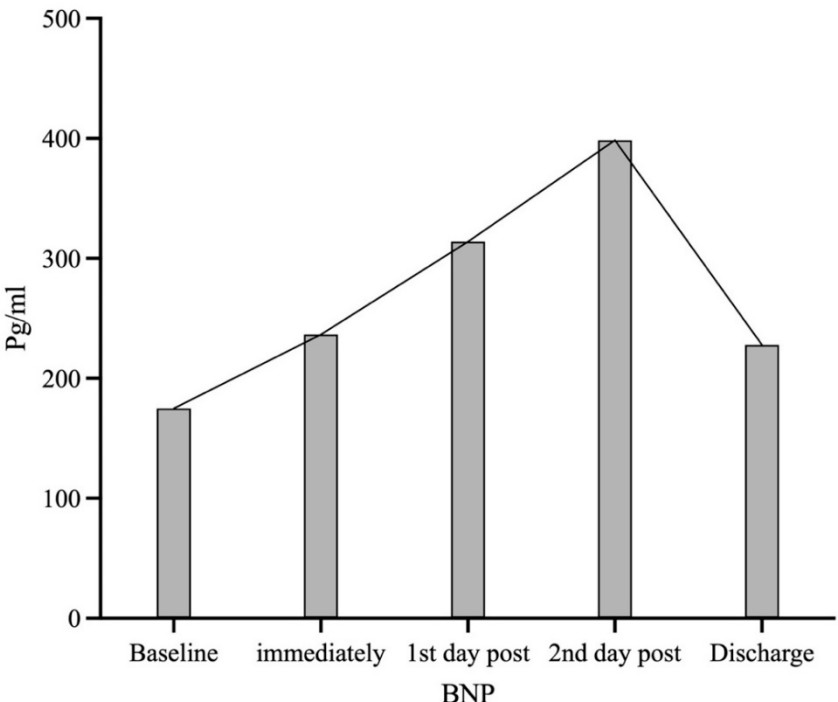

**Figure 3.** The change of brain natriuretic peptide postoperation.

## 4. Discussion

*4.1. Major Findings of Change in Blood Pressure and Serum Electrolytes*

In our study, we found that SBP increased immediately after the operation and then returned to baseline on the 1st postoperative day; although it was slightly lower than the baseline, there was no significant difference. However, a significant decrease in the SBP level was observed at the discharge compared with the baseline. Meanwhile, after

a short decrease on the 1st and 2nd day postoperation, the DBP level had returned to the baseline level at discharge. Although this research lacks the long-term follow-up results of the blood pressure, the hospitalization outcomes of patients undergoing the E-clip procedure that we observed are familiar with the previous studies on BP changes in the patients following other epicardial LAA closure [8,10] and provide a more specific change of the blood pressure during the hospitalization. Maybrook et al. [8] reported that LAA exclusion via the LARIAT suture-delivery device results in an early and persistent decrease in systolic BP. Maybrook et al. [8] observed a significant reduction in systolic BP (mmHg) at 24 h (113.3 ± 16.0; $p < 0.0001$) and 72 h (119.0 ± 18.4 mmHg; $p < 0.0001$) post-LARIAT when compared with pre-LARIAT BP (138.2 ± 21.3). The decrease in systolic BP persisted after a 6-month follow-up (128.8 ± 17.3; $p = 0.0005$). Turagam et al. [11] observed a significant decrease in SBP at 3-month follow-up, and it is worth noting that compared to the baseline level (137.50 mmHg), SBP was significantly decreased immediately after operation (115.40 mmHg) in the study that included 38 patients who underwent epicardial LAA closure on blood pressure change following epicardial LAA exclusion. We noticed that in previous studies, baseline blood pressure was always measured 3 weeks before the procedure in a clinic setting, whereas we measured it 1–2 days before the procedure. Given that we generally adjust antihypertensive drugs in patients with hypertension before the procedure, we hypothesized that the difference in blood pressure change between our study and the previous studies was caused by the difference in baseline blood pressure collection time points. Furthermore, all of the patients in our study had suffered an ischemic stroke a few months before the procedure, with a significant increase in SBP observed immediately after the ischemic stroke in previous studies; although this increase would disappear in the next few weeks, the effect of the stroke on SBP still needed to be considered.

In our study, most patients experienced an acute decrease in serum sodium, which was also observed in the previous studies [8,10,11]. Remarkably, Holmes, et al. [10] discovered that among patients with AF after epicardial LAA closure by LARIAT, the average serum sodium level decreased by 4.98 ± 3.74 mmol/L within 48 h postprocedure; there were 32 patients (52.4%) decreased ≥4 mmol/L, and 6 patients (9.8%) decreased ≥10 mmol/L, but almost all of them returned to the normal level before discharge. In our study, a more significant decrease was observed: the average serum sodium level decreased by 4.72 ± 2.80 mmol/L within 48 h postprocedure, with 25 patients (48.08%) decreasing by ≥4 mmol/L, and 3 patients (5.77%) decreasing by ≥10 mmol/L. Furthermore, a significant difference in serum sodium was still observed between discharge and baseline. All of the patients in our study had an ischemic stroke which could be the cause of the persistent decrease in serum sodium. However, more research is needed to determine the precise mechanism underlying this phenomenon. Hyponatremia is a common post-operative complication that can cause severe central nervous system dysfunction such as headache, nausea, lethargy, disorientation, or depressed reflexes [12]. In addition, cerebral edema caused by rapid and significant fluctuations in serum sodium levels can result in serious complications such as seizures, coma, brain damage, and brain-stem herniation [12]. We observed a predictable decline in serum sodium as a result of stand-alone thoracoscopic LAA clipping, which could explain the early, unexplained postoperative serum sodium decline and avoid further costly and unnecessary investigation. Our findings suggest that for patients with pre-existing hyponatremia or who are on diuretics, postoperative fluid management should be appropriate or diuretics should be reduced; otherwise, patients' hospital stays may be prolonged.

### 4.2. Proposed Hypothesis for the Current Observations

LAA is the derivative of the primitive fetal atrium; in addition to the known electrical and mechanical/reservoir properties [4,13,14], recent research has revealed that LAA has important neurohormonal function [4,6,7]. LAA is a well-known source of ANP, with approximately 30% of total ANP stored within ANP-producing granulocytes in the LAA. ANP and BNP secretion is stimulated by a variety of factors, the most sensitive of which is

distention in the LAA wall [15,16]. ANP and BNP, which are secreted into the circulation from the heart, produce diuretic, natriuretic, and hypotensive activity by affecting the vessel and kidney and by functionally antagonizing the renin–angiotensin–aldosterone system (RAAS) [17,18]. Increases in ANP and BNP induce natriuresis and diuresis in patients with heart failure, resulting in the normalization of sodium levels and blood pressure. In the absence of heart failure, however, serum ANP and BNP elevation can cause hyponatremia and unwanted hypotension [19].

Changes in RAAS caused by changes in ANP and BNP following epicardial LAAC (left atrial appendage clipping) are currently thought to cause fluctuations in homeostasis. However, the precise changes in ANP and BNP postoperatively, as well as the actual mechanism of perioperative changes in homeostasis, remain unknown. Although only BNP was measured in our study and revealed a trend that was familiar from the previous studies [20–22], a significant increase in BNP level was observed postoperatively, which persisted to the discharge and indicated a possible reason for the change of BPs and electrolytes we observed following the procedure.

### 4.3. Study Limitations

The study has several significant limitations. This is a single-center study, which has all of the limitations that come with retrospective and observational studies. The sample size is also a limitation of the study. First and foremost, there is a lack of long-term follow-up data, making it difficult to determine whether the aforementioned changes are long-term or temporary. Furthermore, the level of ANP and the expression of the RAAS system were not measured directly and in real time. As a result, we can only form a hypothesis and infer the relationship based on the information we have. Besides, the changes mentioned above about electrolytes and hemodynamics are susceptible to diuretics use, fluid management, and psychological factors in patients. Thus, more research is required to precisely explain the further mechanisms.

### 5. Conclusions

Stand-alone thoracoscopic left atrial appendage clipping causes an acute decrease in serum sodium and systolic blood pressure. These findings have important clinical implications as they suggest that surgeons should pay more attention to a patient's electrolyte and fluid levels following the epicardial LAA clipping, as well as provide early data for the studies on postprocedure neurohormone and RAAS-related changes secondary to epicardial LAA clipping. Furthermore, the significant decrease in patients' systolic blood pressure at discharge indicates that AF patients with stroke may receive more benefits from stand-alone thoracoscopic LAA clipping. The abovementioned points were hypothesized based on the data we collected; therefore, the accuracy and further mechanisms require further investigation.

**Author Contributions:** Conceptualization, Y.C.; Data curation, X.H. and C.Y.; Formal analysis, Y.C.; Investigation, C.Y.; Methodology, Y.C.; Supervision, D.X.; Writing—original draft, Y.C.; Writing—review & editing, D.X. All authors have read and agreed to the published version of the manuscript.

**Funding:** This research received no external funding.

**Institutional Review Board Statement:** This is a retrospective, observational study that was conducted on already available data, and we have checked with the ethical committee of Beijing Tiantan hospital to make sure that ethical approval is not needed in this study and that it complies with the requirements of China.

**Informed Consent Statement:** Not applicable.

**Data Availability Statement:** Not applicable.

**Acknowledgments:** The authors would like to thank Jing Zhao for many useful suggestions in statistical analysis.

**Conflicts of Interest:** The authors declare no conflict of interest.

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
