# Peer review of "The Significant Decrease of Serum Sodium and Blood Pressure following Thoracoscopic Left Atrial Appendage Clipping"

_2813-2475, doi:10.3390/jvd1020011_

Round 1

Reviewer 1 Report

The Authors have nicely written an important clinical aspect. Overall English is very poor to present these important findings. 

I request authors to revise your discussion and conclusion. Moreover, please modify the title to justify the content of this manuscript.

Please add a pilot graph or Algorithm for how you performed this study. The sample size varies from all groups so it's hard to justify the conclusion from the written content.

Author Response

Dear reviewer,

Thank you for taking the time to process the submission of our original paper entitled “The effects of Stand-alone Thoracoscopic left atrial appendage clipping on blood pressure and electrolytes.We appreciate your kind suggestions and we have amended the manuscript accordingly.

  1. Overall English is very poor to present these important findings. 

We sincerely apologize for the inconvenience of reading caused by grammatical problems or typing errors. however, we have asked the professional English editor (https://www.enago.cn ) to correct the issues without altering the paper’s original meaning and the editing certificate would be uploaded if necessary.

  1. I request authors to revise your discussion and conclusion. Moreover, please modify the title to justify the content of this manuscript.

Thanks for your careful reading of our manuscript, we have checked the manuscript again and changed its title to The significant decrease of serum sodium and blood pressure following thoracoscopic left atrial appendage clipping.

  1. Please add a pilot graph or Algorithm for how you performed this study. The sample size varies from all groups so it's hard to justify the conclusion from the written content.

  We sincerely apologize for the errors in the sample size, there was nothing more than we forgot to change the number after we enrolled a new banche of patients.

We appreciate your warm work earnestly and hope that the correction will meet with approval. Once again, thank you very much for your comments and suggestions.

Prof.Dong Xu

Reviewer 2 Report

The current study aimed to investigate the change of electrolytes and blood pressure in AF patients following the thoracoscopic LAA clipping surgery. The study is behind a relevant topic, but some concerns regarding the study's rationale and methods must be addressed.  

Introduction 

This section is short and shallow. Please provide a clear rationale with scientific content that justifies the current study. 

Methods

My primary concern is regarding the study's design. Why did the authors choose a single-arm design? Should be the correct design comparison between intervention by the novel LAA clipping device and the traditional method? If the authors can provide this data the study will be much improved. 

Author Response

Dear reviewer,

Thank you for taking the time to process the submission of our original paper entitled “The effects of Stand-alone Thoracoscopic left atrial appendage clipping on blood pressure and electrolytes.We appreciate your kind suggestions and we have amended the manuscript accordingly.

  1. Introduction: This section is short and shallow. Please provide a clear rationale with scientific content that justifies the current study. 

  Thanks for your careful reading of our manuscript, we have checked the manuscript again and added some content necessary in the introduction to make it more clear.

  1. Method: Why did the authors choose a single-arm design? 

The thoracoscopic LAA clipping was performed by the E-clip which is a novelty device created in the recent year, it’s difficult for us to develop a control group.

  1. Method: Should be the correct design comparison between intervention by the novel LAA clipping device and the traditional method?

   Thank you for pointing this out. Actually, the percutaneous LAA closure through the LARIAT device could induce the In situ necrosis of LAA, as well as the procedure performed by the E-clip (Med-zenith, Beijing) device. We believe that the novel LAA clipping device and the traditional method are comparable since both of these procedures were generally identified as epicardial LAA closure in the previous studies.

We appreciate your warm work earnestly and hope that the correction will meet with approval. Once again, thank you very much for your comments and suggestions.

Prof.Dong Xu

Round 2

Reviewer 2 Report

My all concerns were addressed.